# Mycorrhizas Affect Physiological Performance, Antioxidant System, Photosynthesis, Endogenous Hormones, and Water Content in Cotton under Salt Stress

**DOI:** 10.3390/plants13060805

**Published:** 2024-03-12

**Authors:** De-Jian Zhang, Cui-Ling Tong, Qiong-Shan Wang, Shu Bie

**Affiliations:** 1Key Laboratory of Cotton Biology and Breeding in the Middle Reaches of the Yangtze River, Ministry of Agriculture, Industrial Crops Institute, Hubei Academy of Agricultural Sciences, Wuhan 430064, China; zhangdejian0553@126.com; 2Hubei Key Laboratory of Waterlogging Disaster and Agricultural Use of Wetland, College of Horticulture and Gardening, Yangtze University, Jingzhou 434023, China; 3Jingzhou Institute of Technology, Jingzhou 434020, China

**Keywords:** AMF, antioxidant system, hormones, photosynthesis, saline–alkali stress

## Abstract

Saline–alkali stress seriously endangers the normal growth of cotton (*Gossypium hirsutum*). Arbuscular mycorrhizal fungi (AMF) could enhance salt tolerance by establishing symbiotic relationships with plants. Based on it, a pot experiment was conducted to simulate a salt environment in which cotton was inoculated with *Paraglomus occultum* to explore its effects on the saline–alkali tolerance of cotton. Our results showed that salt stress noticeably decreased cotton seedling growth parameters (such as plant height, number of leaves, dry weight, root system architecture, etc.), while AMF exhibited a remarkable effect on promoting growth. It was noteworthy that AMF significantly mitigated the inhibitory effect of salt on cotton seedlings. However, AMF colonization in root and soil hyphal length were collectively descended via salt stress. With regard to osmotic regulating substances, Pro and MDA values in roots were significantly increased when seedlings were exposed to salt stress, while AMF only partially mitigated these reactions. Salt stress increased ROS levels in the roots of cotton seedlings and enhanced antioxidant enzyme activity (SOD, POD, and CAT), while AMF mitigated the increases in ROS levels but further strengthened antioxidant enzyme activity. AMF inoculation increased the photosynthesis parameters of cotton seedling leaves to varying degrees, while salt stress decreased them dramatically. When inoculated with AMF under a salt stress environment, only partial mitigation of these photosynthesis values was observed. Under saline–alkali stress, AMF improved the leaf fluorescence parameters (φPSII, Fv′/Fm′, and qP) of cotton seedlings, leaf chlorophyll levels, and root endogenous hormones (IAA and BR); promoted the absorption of water; and maintained nitrogen balance, thus alleviating the damage from salt stress on the growth of cotton plants to some extent. In summary, mycorrhizal cotton seedlings may exhibit mechanisms involving root system architecture, the antioxidant system, photosynthesis, leaf fluorescence, endogenous hormones, water content, and nitrogen balance that increase their resistance to saline–alkali environments. This study provide a theoretical basis for further exploring the application of AMF to enhance the salt tolerance of cotton.

## 1. Introduction

Modern agriculture is facing multiple abiotic stresses, such as salt damage, drought, waterlogging, heat, and chilling [1,2,3]. Soil salinity affects nearly 33% of watered land and 20% of cultivable land worldwide, presenting a major problem that limits crop yields [2,4,5]. Saline–alkali stress consistently subjects plants to superionic and hypertonic conditions that cause water deficit and ion toxicity, thereby limiting water and nutrient absorption as well as cell expansion and, consequently, leading to plant death [6]. Water deficit may cause severe physiological drought, decrease transpiration, close stomata, and inhibit the photosynthesis of leaves [7]. Excess Na^+^ and alterations in intracellular Ca^2+^ levels cause ion toxicity, with plants regulating different ion transporters and protein kinases to maintain ion homeostasis [8,9]. Salt stress disrupts the dynamic equilibrium of plants, causing the excessive accumulation of reactive oxygen species (ROS), which is toxic to plants and triggers the activation of the antioxidant enzyme system, a plant’s defense system [10,11]. Moreover, phytohormones also participate in salt stress signal sensing and defense system mediation. The accumulation of ABA in plant cells and locations has been identified as bio-labeling for plant stress responses. Furthermore, the crosstalk between phytohormones also plays a fundamental role in salt stress [9,12,13,14]. Thus, salt stress seriously limits plant growth as well as crop production and quality.

Cotton (*Gossypium hirsutum*), one of the most crucial industrial and fiber crops worldwide, has a wide planting range and easy cultivation [15,16]. China is not only a major global producer of cotton but also a major consumer of cotton. China already accounts for nearly a quarter of the world’s cotton production and a third of global consumption and ranks first in the world in cotton and cotton yarn imports [17]. The ever-growing manufacturing requirements have made growing cotton more challenging. Cotton faces many abiotic stresses (phosphorus and water deficiency, as well as drought, salt, cold, and heat stresses) and some biotic stresses (insect pests and plant diseases) [16,17,18,19,20]. Several studies have shown that soil salinization is an important environmental limiting factor for cotton. It seriously affects its growth and fiber quality, and the sensitivity of cotton to salt stress depends on the growth stage of the crop, the cultivation method used, the duration of stress, and the concentration and type of salt [21,22]. Cotton is classified as a moderately salt-tolerant crop, primarily concerning NaCl stress [23,24]. NaCl stress results in delayed flowering time, fewer fruiting positions, reduced boll weight, deteriorations in fiber quality, reductions in chlorophyll content, damaged photosynthesis, and affected stomata opening, which ultimately affect seed cotton yield [24,25]. Cotton under salt stress produces high levels of ROS and reduces the content of osmotic regulators, which damages the cell structure and disrupts regular plant growth [26,27]. Salt stress influences osmotic regulating substances and the repression of ion homeostasis during the germination of cotton seeds [28]. 

Arbuscular mycorrhizal fungi (AMF) widely exist in soil and establish a mycorrhizal symbiotic relationship with most terrestrial plants, which can markedly enhance the ability of plants to resist biotic and abiotic stresses [29,30]. Inoculation with *Rhizophagus intraradices* has been found to reduce the presence of certain terpenoids, while *Funneliformis mosseae* regulates auxin transport and synthesis levels in the root to improve trifoliate orange growth. Both pathways induce plant defense responses and, ultimately, improve the plant’s ability to drought tolerate [31]. AMF regulates aquaporin expression in host plants to respond to water stress [32]. Under salt stress, AMF inoculation can improve plant growth through increased nutrient absorption, decreased water deficit, and increased photosynthetic efficiency [33,34,35]. It also affects the concentration and profile of organic acids and polyamines, reducing damage to the ultrastructure and enhancing tolerance to salt stress [36,37]. Moreover, AMF increases antioxidant (peroxidase-POD, superoxide dismutase-SOD, catalase-CAT, glutathione reductase-GR, ascorbate peroxidase-APX, and ascorbic acid-ASA) contents and decreases malondialdehyde (MDA) and electrolyte levels in plants to alleviate oxidative stress and membrane damage induced via salinity [38,39]. AMF-inoculated plants exhibit increased levels of fructose and free proline (PRO) and decreased sucrose content, contributing to cell osmotic regulation. This mechanism helps to reduce salt toxicity and sustain membrane and DNA integrity [40,41]. In cotton, utilizing both *Bacillus subtilis* and *Bacillus pumilus* can significantly enhance salt stress tolerance [42].

However, the potential of AMF to improve salt resistance in cotton and the related mechanisms have not been extensively studied. In this study, four groups of cotton (CK-inoculation without *Paraglomus occultum* under non-salt stress, Po-inoculation with *Paraglomus occultum* under non-salt stress, NaCl-inoculation without *Paraglomus occultum* under salt stress, and NaCl + Po-inoculation with *Paraglomus occultum* under salt stress) were conducted to examining plant growth performance, antioxidant enzyme system, photosynthesis, phytohormones, and respiratory metabolism. Additionally, we aim to identify the responses of cotton to salt stress, understand its adaptation mechanisms, and provide more suggestions for improving cotton yield.

## 2. Results

### 2.1. Effects of Salt Stress on AMF Colonization

Soil mycorrhizal fungal hyphae and root mycorrhizal colonization were detected in the AMF seedlings, but not in the non-AMF seedlings (Figure 1 and Table 1). Hence, the roots of cotton established a symbiotic relationship with *Paraglomus occultum* (Po). Under CK treatment, root mycorrhizal colonization was 77.4%, while it was reduced to 52.32% under salt stress (NaCl + Po treatment). In addition, salt stress (NaCl + Po treatment) also significantly decreased soil hyphal length by 28.43% compared with Po treatment. This indicates that the infection activity of AMF was inhibited via salt stress.

### 2.2. Effects of Salt Stress and AMF on Plant Growth and Root System Architecture

As shown in Figure 2, salt stress suppressed seedling growth, while AMF promoted it significantly. Plant height, stem diameter, leaf number, and shoot and root dry weights were markedly decreased in salt-treated plants compared to non-salt-treated ones, irrespective of AMF status (Table 1). Under salt stress, plant height, stem diameter, leaf number, and shoot and root dry weights were increased by 26.3%, 15.1%, 32.0%, 74.2%, and 11.4%, respectively, with AMF treatments (Table 1). However, AMF inoculation significantly enhanced plant height by 22.7%, stem diameter by 13.6%, leaf number by 10.1%, and shoot and root dry weights by 70.6% and 15.2%, respectively, compared to non-AMF treatment under non-salt stress.

Additionally, AMF and salt stress exerted noticeable effects on the root system architecture (Figure 3). The mycorrhizal seedlings showed a strong root system architecture, which weakened under salt stress treatment. As shown in Table 2, AMF inoculation dramatically enhanced total root length by 23.3% and 13.6% compared to non-AMF treatment under non-salt stress and salt stress conditions. Furthermore, salt stress significantly reduced total root length by 7.7% and 19.8% compared to non-salt treatment under both non-AMF and AMF inoculation conditions (Table 2). However, AMF had no observable effect on root projected area, total root surface area, root average diameter, and root volume, regardless of the presence of salt stress (Table 2).

In these datasets, saline–alkali stress suppressed plant growth, resulting in weakened root configuration, while AMF relieved this inhibitory effect.

### 2.3. Effects of Salt Stress and AMF on Osmotic Regulating Substances

By measuring the contents of osmotic regulating substances, we found that salt stress and AMF do not affect the contents of soluble protein and soluble sugar in cotton plant roots after the onset of salinity stress (Table 3). With regard to Pro and MDA, salt stress significantly increased their contents by 96.2% and 99.2% compared to the control group under non-AMF inoculation, while increases of 44.0% and 64.1% were observed under conditions with AMF inoculation (Table 3). Although AMF had no effect on the contents of Pro and MDA under non-salt stress conditions, this was reduced by 29.3% and 19.9% compared to the control group under salt stress (Table 3). Thus, saline–alkali stress increased the contents of some osmotic regulating substances, while AMF mitigated these effects.

### 2.4. Effects of Salt Stress and AMF on H_2_O_2_ and Antioxidant Enzyme Activity

Similar effects were gained in DAB and NBT staining experiments, showing no obvious changes in H_2_O_2_ content in cotton roots when treated with AMF under non-salt stress (Table 4). However, AMF decreased this by 15.1% under salt stress (Table 4). Compared to non-AMF inoculation, the activities of CAT, POD, and SOD in cotton roots treated with AMF were significantly increased, regardless of the presence of salt stress (Table 4). Compared with CK, NaCl treatment significantly increased the activities of SOD, POD, and CAT by 101.0%, 166.4%, and 182.4%, which can be further increased to 129.0%, 228.0%, and 261.8% when inoculated with AMF under salt stress (Table 4). Thus, salt stress increased ROS levels and enhanced antioxidant enzyme activity, while AMF inoculation mitigated the increases in ROS levels but further strengthened antioxidant enzyme activity.

### 2.5. Effects of Salt Stress and AMF on Photosynthesis Parameters in Leaves

As shown in Table 5, AMF inoculation increased the net photosynthetic rate (Pn, 12.83%), stomatal conductance (Gs, 13.45%), intercellular carbon dioxide concentration (Ci, 12.77%), and transpiration rate (Tr, 6.79%) of cotton without salt stress. Applying salt stress resulted in reduces in the Pn (27.61%), Gs (11.69%), Ci (31.67%), and Tr (17.98%). Under NaCl + Po treatment, although NaCl stress led to significant decreases in Pn (21.45%), Gs (11.34%), Ci (31.99%), and Tr (12.53%) compared to Po treatment, Gs and Tr levels were similar to those of CK. This indicates that AMF may promote the growth of cotton by regulating these two pathways (Gs and Tr) under saline–alkali stress.

### 2.6. Effects of Salt Stress and AMF on Leaf Fluorescence Parameters

The effects of salt stress and AMF inoculation on the fluorescence parameters of cotton leaves were different (Table 6). Salt stress significantly reduced the efficiency values of photosystem II (φPSII), maximal photosystem II quantum yield (Fv′/Fm′), and photochemical quenching coefficient (qP) but noticeably increased the non-photochemical quenching (NPQ) value, regardless of the presence of AMF inoculation. Compared with CK, AMF inoculation markedly increased φPSII (31.74%), Fv′/Fm′ (13.95%), and qP (25.77%) values while dramatically decreasing the NPQ (34.32%) value of cotton without salt stress (Table 6). Similarly, the values of φPSII, Fv′/Fm′, and qP under NaCl + Po were still significantly higher than those under NaCl (14.81%, 9.1%, and 21.52%), while the value of NPQ showed a significant decrease (22.73%). However, all values for the leaf fluorescence parameters under NaCl + Po were similar to those of CK. In short, AMF likely promotes the growth of cotton by regulating leaf fluorescence parameters under salt stress.

### 2.7. Effects of Salt Stress and AMF on Root Endogenous Hormone Levels

The salt treatment caused an obvious decrease in the IAA and BR concentrations in cotton roots but increased ABA levels, while IAA and BR levels were significantly increased and ABA concentration was decreased after inoculation with AM fungi (Table 7). However, GA levels in cotton roots showed no significant differences across all treatments. Compared with CK treatment, IAA and BR levels under Po treatment increased by 18.14% and 17.43%, respectively, while they decreased by 10.81% and 30.71%, respectively, after treatment with NaCl (Table 7). Compared with NaCl treatment, IAA, BR, and GA concentrations in the roots under NaCl + Po treatment were increased by 19.22%, 25.15%, and 1.80%, respectively (Table 7). Therefore, IAA, BR, and ABA concentrations in cotton roots are significantly affected by AMF inoculation and salt stress. This suggests that AMF inoculation in a salt environment plays an important role in regulating the synthesis of endogenous hormones in plants.

### 2.8. Effects of Salt Stress and AMF on Water Content, Chlorophyll Levels, and Nitrogen Balance of Leaves

The fresh weight of plants is affected by water content; thus, the water content and water saturation deficit of leaves were further investigated. Under the influence of salt stress, the water content and water saturation deficit decreased by 4.55% and 19.28% but returned to CK levels after inoculation with AMF, likely because AMF has positive effects on water uptake under a salt stress environment (Table 8). Compared with CK, chlorophyll levels and nitrogen balance under NaCl treatment significantly decreased by 15.78% and 24.91% but returned to CK levels after inoculation with AMF, indicating that AMF plays an important role in promoting chlorophyll synthesis and nitrogen balance (Table 8). However, the levels of flavonoids in cotton leaves showed no significant differences across all treatments.

## 3. Discussion

Salt stress endangers the normal growth of cotton seriously [22,25]. Sessile plants have developed variable mechanisms to respond to salt stress signals and accommodate high superionic and hypertonic conditions [9,43]. AMF widely exists in soil, and some fungi improve host plant salt stress tolerance by forming a symbiotic relationship with plant root systems, being one of the most important plant mechanisms that allow plants to adapt to salt stress [38,44]. Our results indicated that inoculating cotton with *Paraglomus occultum* improved its tolerance to saline–alkaline conditions and exhibited multiple positive effects.

Under salt stress, the infection activity of AMF and plant physiology were inhibited [45]. In this study, *P. occultum* was successfully colonized with cotton. AMF colonization increased cotton growth in terms of both root and shoot biomass compared to non-AMF-inoculated plants. Although plant biomass was decreased via salt concentrations, it was relieved by symbiosis with AMF. This is consistent with the results of previous studies on *Solanum lycopersicum*, *Lactuca sativa*, rice, and *Arundo donax* L. [46,47,48,49]. In addition, the root mycorrhizal colonization rate and soil hyphal length of *P. occultum* were significantly decreased via salt treatment. As Juniper et al. [50] reported, AMF hyphal growth and spore germination were limited by salinity. Thus, salt stress inhibits the infection activity of *P. occultum*, suggesting that *P. occultum* may be unsuitable for saline stress.

Roots sense and respond to environmental signals and, thus, could be used to reflect the conditions of root growth [51,52]. Here, our analysis of the root system architecture results showed that AMF inoculation and salt treatment significantly changed root length. However, AM fungi increased the average root diameter, root volume, and root tip, which helps plant roots absorb more water and nutrients for better growth [53,54]. The promoting effect of mycorrhiza on plant roots decreased by 9.7% under salt stress, and the inhibitory effect of salt stress on root length increased by 12.1% under mycorrhizal conditions. Symbiosis with AMF indeed relieved salt stress. This further supports the conclusion that mycorrhizal plants exhibit elevated salt tolerance [55].

Salt stress improved intracellular osmotic pressure and could cause cell membrane injury. It is well known that the content of MDA reflects the degree of cell damage caused by reactive oxygen species and membrane lipids [56]. Proline (Pro) is a substance that regulates osmotic activity. In our study, the contents of MDA and Pro under the salt stress treatments were significantly higher than those under the non-salt stress treatments, with AMF mitigating these effects. However, the contents of soluble protein and soluble sugar were unaffected under the two treatments. This demonstrates that salt stress damagess the cell membrane in leaves, leading to the outflow of osmotic substances and, thus, increases in MDA and Pro levels, with AMF relieving these effects. Zong et al. [39] verified this hypothesis and found that the impact of AMF became significantly noticeable at a concentration above 160 mM NaCl. However, there are contradictions regarding how AM affects the accumulation of proline. On the one hand, under salt stress, the proline content was lower with AMF treatment. This indicates that salt stress increases the rate of proline metabolism, thus resulting in proline overproduction. Similar to our study, AMF primarily relieves this effect [36,57,58]. On the other hand, AMF stimulates proline increases, suggesting that AMF plays a dominant role in the osmotic regulation capacity of plants [59,60].

Salinity disrupts the dynamic equilibrium of plants and causes the leakage of electrolytes [8,9], thereby leading to increased peroxidation of membrane lipids and, thus, the excessive accumulation of ROS [39,61], which are toxic to plants and trigger the activation of the antioxidant enzyme system [11]. Many studies have reported that AMF can positively regulate antioxidant enzymes to alleviate oxidative and membrane damage induced via salinity [45,62]. In our study, CAT, POD, and SOD activities in AM-inoculated plants increased significantly under salinity conditions. H_2_O_2_ is a byproduct of oxygen metabolism that has the potential to be harmful [45,63]. Additionally, the decrease in H_2_O_2_ content in the AMF group further verifies that AMF symbiosis helps plants induce greater antioxidant enzyme production to combat oxidative damage. These results are consistent with previous studies [60,64].

Saline–alkali stress causes ion imbalance, thus limiting water absorption. Plants respond by closing stomata to mitigate water deficit and single ion toxicity, thereby inhibiting leaf photosynthesis [7,65]. However, this prevents the production of organic matter on chloroplasts and, in severe cases, leads to plant death [66]. It has been described that photosynthesizing is a crucial indicator of plant stress tolerance [39]. In rice, Turan et al. [67] found that salt stress inhibits chlorophyll biosynthesis intermediates to reduce the accumulation of chlorophyll. In addition, AMF colonization relieves the damage to photosynthesis performance under stress by regulating photosynthetic activity (Pn, Ci, and Tr), water content, and stomatal conductance (Gs) [6,68]. In this study, the Gs, Pn, Ci, and Tr levels of the AMF group were dramatically higher than those of the non-AMF group under salt stress, similar to water content, water saturation deficit, and chlorophyll content. It can be inferred that 150 mmol NaCl disrupted the mechanism of photosynthesis. AMF colonization alleviates this stress by promoting plant root elongation, which expands the water absorption area of plants to relieve water shortage in plant cells, thus inhibiting stomatal closure, reducing chloroplast damage, and improving the efficiency of light energy utilization. This is consistent with previous studies on black locust, *Xanthoceras sorbifolium*, and *Populus simonii* [6,39,69].

The primary activity of photosystem II (PSII) is to sense surrounding factors, measured non-invasively via chlorophyll fluorescence, thus being an effective indicator of plant photosynthesis under various stresses [70]. Liang et al. [71] summarized that salt stress will disturb the electron transport chain and produce excessive ROS to damage the plasma membrane. In Oryza sativa, Porcel et al. [72] reported that AMF-colonized plants down-regulate φPSII and qP but up-regulate NPQ values under salt stress and that Fv′/Fm′ reflects PSII transformation efficiencies, indicating the capacity for salt tolerance [73]. In this work, salt stress dramatically decreaseed the values of φPSII, Fv′/Fm′, and qP but noticeably increased the NPQ value, while inoculation with AMF mitigated these phenomena. Thus, the conjecture that AMF inoculation alleviates damage to the PSII system and promotes the growth of cotton under salt stress is consistent with the findings of Wu et al. [74] and Jia et al. [75].

Phytohormones also participate in salt stress signal sensing and defense system mediation [13,14]. Herein, we determined the endogenous hormone levels (IAA, BR, GA, and ABA). Numerous researchers have found that IAA plays an important role in enhancing salt stress tolerance in different crops [76,77,78]. Brassinosteroids (BRs) crosstalk with polyamines (PAs) to respond to salt stress, leading to an increase in spermine concentration in leaves [79]. Many observed aspects of crop phytohormone crosstalk and AM symbiosis accelerate plant stress tolerance [80,81]. Our study showed that root IAA and BR levels were significantly reduced under salt stress and that colonization with AMF inhibited these decreases. This is consistent with previous research on trifoliate orange, tomato, and melatonin-treated cotton [78,80,81]. Furthermore, only IAA and BR concentrations in cotton roots significantly changed, implying that AMF colonization increases cotton salt tolerance primarily through interactions between auxin and Brassinosteroids.

## 4. Materials and Methods

### 4.1. Experimental Design

The experiment was arranged in a 2^2^ factorial completely randomized blocked design: inoculation with or without *Paraglomus occultum* and salt stress treatments with 150 mmol·L^−1^ NaCl (CK-inoculation without *Paraglomus occultum* under non-salt stress, Po-inoculation with *Paraglomus occultum* under non-salt stress, NaCl-inoculation without *Paraglomus occultum* under salt stress, and NaCl + Po-inoculation with *Paraglomus occultum* under salt stress). According to Sun et al. [82] and Chen et al. [83], 150 mmol·L^−1^ concentration is the salt stress level in cotton plants. In addition, we did a concentration gradient test in the early stage, through which we learned that the growth and development of cotton were significantly inhibited under 150 mmol·L^−1^ NaCl. Each treatment was replicated 5 times, and each replicate had 5 pots, and each pot had 9 seedlings. So that each treatment had 225 seedlings.

### 4.2. Plant Culture

AMF (*Paraglomus occultum*) was provided by Prof. Wu, who come from Yangtze University, and propagated with white clover (*Trifolium repens*) containing spores, mycorrhizal hyphae, and infected root segments. Pre-experimental represented cotton was positively impacted by the presence of this fungal strain.

The tested cotton seeds were ZS08 cotton, provided by the Key Laboratory of Cotton Biology and Breeding in the Middle Reaches of the Yangtze River. The seeds were sterilized with 70% alcohol solutions for 15 min, followed by washing with sterile water, and then placed in river sand in an autoclave (121 °C, 0.1 MPa, 1 h) under the conditions of 28/20 °C day/night temperature, 16/8 h light/dark cycle, and 80% relative air humidity for germination. Once seedlings grew two leaves of uniform size, they were selected for transplanting into a pot (upper diameter: 23 cm, lower diameter: 20.5 cm, high: 17 cm) filled with 6 kg of river sands. The sand was collected from the Yangtze River side and autoclaved at 0.11 MPa, 121 °C for 2 h to eliminate spores of indigenous arbuscular mycorrhizal fungi. For transplanting, we adopted the stratified inoculation method, with 200 g of mycorrhizal inoculum and 5.8 kg sterilized river sands in each AMF treatment group and 6 kg sterilized river sand in the non-AMF-inoculated treatment group.

The experiment was conducted in April 2022 at the Institute of Root Biology, Yangtze University, and continued for 16 weeks. All AMF- and non-AMF-inoculated seedlings were cultivated in a greenhouse that maintained a photosynthetic photon flux density of 400 μmol·m^−2^·s^−1^, with 16 h of light followed by 8 h of darkness, a day/night temperature of 28/20 °C, and a relative air humidity of 60–70%. Every three days, seedlings were irrigated with the hoagland solution (200 mL). Pots were moved every week to prevent any potential interference caused by environmental factors.

When cotton seedlings grew four leaves after transplanting, salt stress treatments began, with NaCl added to Hoagland solutions. The salt stress treatment pot was irrigated with 200 mL of Hoagland solutions containing 150 mmol·L^−1^ NaCl at 3-day intervals. The salt stress treatments lasted for 8 weeks, revealing different plant growth characteristics among the four treatments by that time.

The composition of the basic Hoagland solution [77,84] was as follows: 4.00 mmol/L (mM) Ca (NO_3_)_2_·4H_2_O, 6.00 mM KNO_3_, 2.00 mM MgSO_4_·7H_2_O, 1.00 mM NH_4_H_2_PO_4_, 46.00 μM H_3_BO_3_, 9.20 μmol/L (μM) MnCl_2_·4H_2_O, 0.77 μM ZnSO_4_·7H_2_O, 0.32 μM CuSO_4_·3H_2_O, 0.12 μM H_2_MoO_4_, and 50 mM EDTA-Fe, with a pH range of 5.5–6.00.

### 4.3. Variable Determinations

Plant height was recorded with a ruler, stem diameter was recorded with a vernier caliper, and the number of leaves was manually recorded before harvest. During the harvesting process, the seedlings, categorized into four groups (CK, Po, NaCl, and NaCl + Po), were separated into shoot and root portions, and the biomass of each was measured.

The entire root system was gathered for scanning via the Epson Perfection V700 Photo Dual Lens System (J221A, Epson Co., Ltd., Tokyo, Japan), then using WinRHIZO software (2007b, Regent Instruments Inc., Ottawa, QC, Canada) to examine the morphological traits. The leaves and roots were collected and frozen at −80 °C for the following biochemical factor analysis.

When harvested the seedlings, 1–2 cm root segments from the root tip were cut, stored in an FAA fixing solution, and heated in an oven at 95 °C with immersion in 10% KOH solution. After rinsing with pure water, they were bleached with 10% H_2_O_2_, acidified with 0.2 mol L^−1^ HCl, and stained with 0.05% trimethylene blue lactol solution (VAM). The root mycorrhiza was observed using a binocular biomicroscope (NE610, Ningbo, China). The mycorrhizal colonization percentage was calculated as a percentage of mycorrhiza-colonized root lengths compared to the total observed root lengths [85].

Soil hyphal length was measured utilizing the method of Bethlenfalvay et al. [86]. Here, a 0.5 g soil sample was weighed and mixed with 6 mL of 0.1 mol/L phosphate buffer. Then, 0.8 mL of the mixed solution was combined with 0.4 mL of 0.05% trypan blue solution, heated at 70 °C for 20 min, and then microscopically examined after cooling.

A 10% plant homogenate was prepared to measure osmotic regulating substances, antioxidant enzyme activity, and H_2_O_2_ content using commercial assay kits (Nanjing Jiancheng Technology Co., Ltd., Nanjing, China) in accordance with the manufacturer’s instructions. The activities of CAT, POD, and SOD in cotton leaves were measured according to the Guaiacol method, NBT photochemical reduction method, and UV absorption method. The contents of H_2_O_2_, Pro, and MDA were measured according to the ammonium molybdate method [51], sulfur barbituric acid method [87], and ninhydrin color development method [88]. Soluble protein and soluble sugar were determined according to the anthrone colorimetry and Coomassie brilliant blue (G-250) methods described by Wu et al. [89].

The functional leaves (the 4–5th leaves) of treated cotton were selected to determine leaf photosynthesis parameters, including Pn, Gs, Ci, andTr, using a portable photosynthetic system analyzer (Li-6400, Li-Cor, Inc., Rochester, State of New York, USA), based on the protocol by Fan et al. [90].

Leaf fluorescence parameters, including PSII reaction center actual photochemical efficiency (φPSII), PSII effective light quantum yield (Fv′/Fm′), photochemical quenching coefficient (qP), and non-photochemical quenching coefficient (NPQ), were determined using a luminoscope (Handy-PEA, Lufthansa, England), following the protocol described by Baker et al. [91].

Liquid chromatography–mass spectrometry (LC-MS), based on the protocol of Kojima and Sakakibara [92], was utilized to analyze the extraction of root endogenous hormones, including indole acetic acid (IAA), abscisic acid (ABA), gibberellin (GA), and brassinosteroids (BR), according to the protocol of Hu et al. [78].

Relative water content (RWC) and water saturation deficit (WSD) were determined using the method of Chen et al. [93]. At harvest, 5 plants were randomly selected from each treatment, fresh leaf samples were taken, and their fresh weight (W1) was measured. The sample was then immersed in distilled water for 24 h to measure its saturation weight (W2); finally, it was dried in an oven at 100 °C to constant weight, and the dry weight (W3) was then determined. RWC and WSD are calculated using the following formula.
RWC (%) = (W1 − W3)/(W2 − W3) × 100 
WSD (%) = (W2 − W1)/(W2 − W3) × 100 

Mature functional leaves were selected to determine the contents of chlorophyll, flavonoid, and nitrogen balance index ices (NBI) using a multifunctional blade measuring instrument (Dualex Scientific^+^, French National Academy of Sciences, produced by PESSL, Austria). The absorbance of the UV band was used to measure flavonoid content (Flav), and the light transmittance of two near-infrared bands (710 nm and 850 nm) was used to measure chlorophyll content (Chl). The ratio of chlorophyll content to flavonoid content (Chl/Flav) represented the NBI (an important indicator of crop growth status) [94].

### 4.4. Analysis of Data

All experiments were repeated 5 times. All results are expressed as the mean ± standard error (SE) in tables and figures. The data were statistically analyzed for variance (ANOVA) with SPSS software (Version 26.0, SPSS Inc., Chicago, IL, USA). Duncan’s multiple range tests were used to compare significant differences among treatments at *p* < 0.05.

## 5. Conclusions

Here, a pot experiment was conducted to simulate a saline–alkali environment in which cotton plants were inoculated with *Paraglomus occultum* to explore its effects on the salti tolerance of cotton. Our results showed that salt stress significantly inhibited cotton seedling growth, while AMF exhibited a remarkable effect on promoting growth. It was noteworthy that AMF significantly mitigated the inhibitory effect of salt on cotton seedlings. In this study, mycorrhizal cotton seedlings may exhibit mechanisms involving root system architecture, the antioxidant system, photosynthesis, leaf fluorescence, endogenous hormones, water content, and nitrogen balance that increase their resistance to saline–alkali environments. Our study provide a theoretical basis for further exploring the application of AMF to heighten the salt tolerance of cotton. However, more studies and the potential application of field inoculation are needed.

## Figures and Tables

**Figure 1 plants-13-00805-f001:**
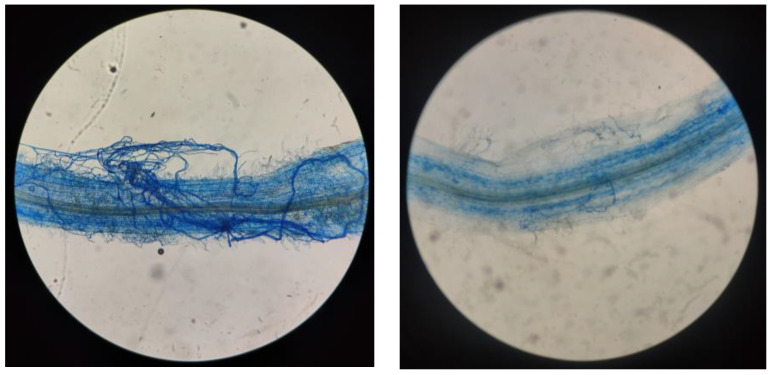
Root colonization of cotton seedlings treated with AMF under salt stress.

**Figure 2 plants-13-00805-f002:**
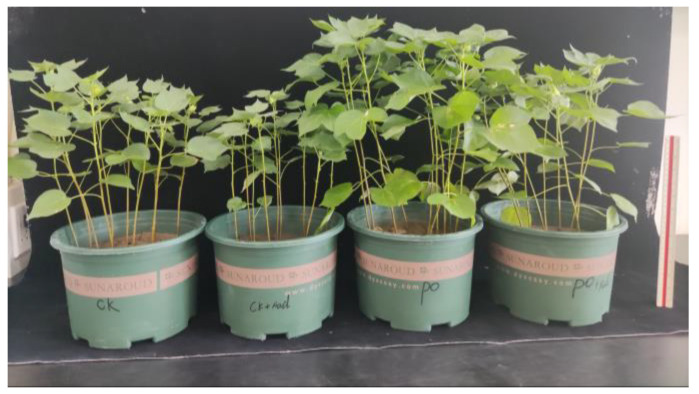
Whole plant morphology in cotton seedlings treated with AMF under salt stress.

**Figure 3 plants-13-00805-f003:**
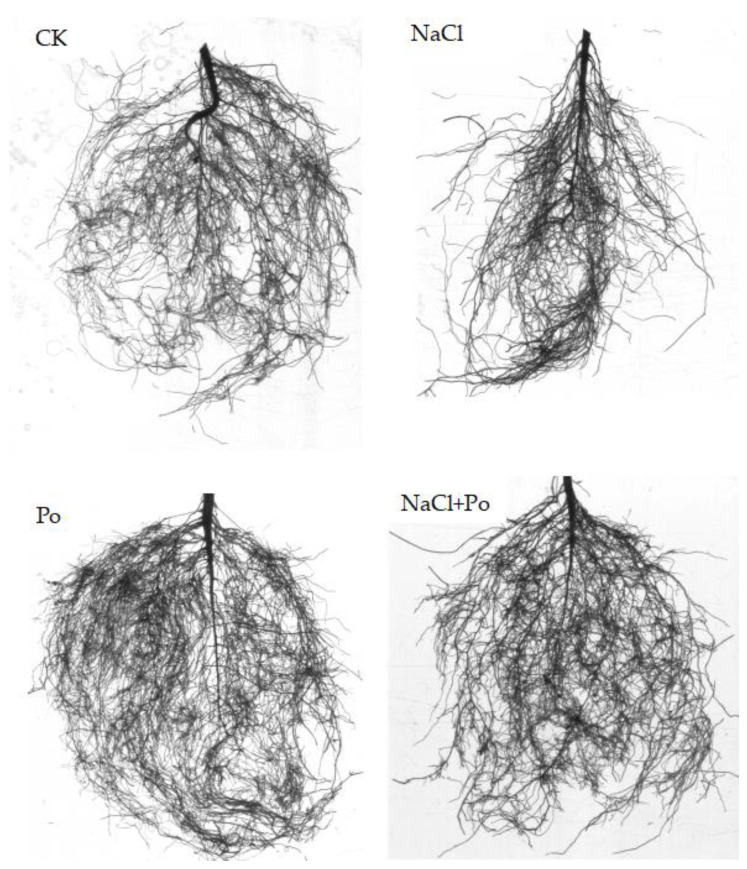
Root system architecture of cotton seedlings treated with AMF under salt stress.

**Table 1 plants-13-00805-t001:** Effects of AMF on root mycorrhizal colonization and plant growth performance of cotton seedlings under salt stress.

Treatments	Plant Height (cm/Plant)	Stem Diameter (mm)	Leaf Number (#/Plant)	Shoot Dry Weight (g FW/Plant)	Root Dry Weight (g FW/Plant)	Mycorrhizal Colonization (%)	Soil Hyphal Length (cm/g Soil)
CK	32.04 ± 0.73 b	2.79 ± 0.06 b	6.75 ± 0.29 b	4.59 ± 0.35 c	2.89 ± 0.25 b	-	-
Po	39.25 ± 0.55 a	3.17 ± 0.08 a	7.43 ± 0.21 a	7.83 ± 0.51 a	3.33 ± 0.20 a	77.41% ± 6.20 a	67.85 ± 5.20 a
NaCl	27.06 ± 0.81 c	2.45 ± 0.06 c	4.81 ± 0.25 c	2.98 ± 0.19 d	2.45 ± 0.18 c	-	-
NaCl + Po	34.11 ± 0.45 b	2.82 ± 0.05 b	6.35 ± 0.19 b	5.19 ± 0.32 b	2.73 ± 0.21 b	52.32% ± 4.20 b	48.56 ± 4.20 b

Data (mean ± SD, *n* = 5) followed by different letters in the column indicate significant differences (*p* < 0.05) between treatments. -: no presence for this variable.

**Table 2 plants-13-00805-t002:** Effects of AMF on root system architecture of cotton seedlings under salt stress.

Treatments	Total Root Length (cm/Plant)	Root Projected Area (cm^2^/Plant)	Total Root Surface Area (cm^2^/Plant)	Root Average Diameter (mm/Plant)	Root Volume (cm^3^/Plant)
CK	221.79 ± 7.47 b	13.07 ± 0.30 ab	16.98 ± 1.45 ab	0.48 ± 0.02 a	2.22 ± 0.20 ab
Po	288.12 ± 9.21 a	13.21 ± 0.27 a	17.50 ± 1.35 a	0.49 ± 0.01 a	2.34 ± 0.19 a
NaCl	204.09 ± 8.12 c	12.84 ± 0.37 b	16.68 ± 1.27 b	0.41 ± 0.01 a	2.16 ± 0.11 b
NaCl + Po	231.81 ± 8.91 b	13.09 ± 0.21 ab	17.11 ± 1.43 a	0.47 ± 0.01 a	2.25 ± 0.10 ab

Data (mean ± SD, *n* = 5) followed by different letters in the column indicate significant differences (*p* < 0.05) between treatments.

**Table 3 plants-13-00805-t003:** Effects of AMF on osmotic regulating substances in roots of cotton seedlings under salt stress.

Treatments	Pro (ug/g)	MDA (nmol/Prot)	Soluble Protein (mg/g)	Soluble Sugar (mg/g)
CK	34.89 ± 1.09 c	502.53 ± 39.44 c	0.85 ± 0.07 a	2.21 ± 0.17 a
Po	33.59 ± 2.35 c	488.54 ± 30.32 c	0.91 ± 0.04 a	2.42 ± 0.14 a
NaCl	68.46 ± 5.50 a	1000.35 ± 60.98 a	0.81 ± 0.05 a	2.03 ± 0.19 a
NaCl + Po	48.37 ± 2.39 b	801.44 ± 51.42 b	0.83 ± 0.05 a	2.26 ± 0.15 a

Data (mean ± SD, *n* = 5) followed by different letters in the column indicate significant differences (*p* < 0.05) between treatments.

**Table 4 plants-13-00805-t004:** Effects of AMF inoculation on H_2_O_2_ and antioxidant enzyme activity in cotton seedling roots under salt stress.

Treatments	H_2_O_2_ (U/Prot)	SOD (U/Prot)	POD (U/Prot)	CAT (U/Prot)
CK	23.95 ± 2.03 c	100.04 ± 9.71 d	30.04 ± 2.45 d	10.71 ± 1.02 d
Po	23.01 ± 2.15 c	130.03 ± 8.79 c	41.73 ± 3.71 c	21.47 ± 1.12 c
NaCl	33.07 ± 2.62 a	201.83 ± 18.21 b	80.03 ± 7.12 b	30.24 ± 2.02 b
NaCl + Po	28.09 ± 1.87 b	229.03 ± 20.46 a	98.41 ± 9.04 a	38.72 ± 3.07 a

Data (mean ± SD, *n* = 5) followed by different letters in the column indicate significant differences (*p* < 0.05) between treatments.

**Table 5 plants-13-00805-t005:** Effects of AMF on photosynthesis parameters of cotton seedling leaves under salt stress.

Treatments	Pn (μmol/m^2^·s)	Gs (μmol/m^2^·s)	Ci (μmol/mol)	Tr (mmol/m^2^·s)
CK	23.22 ± 0.25 b	1.71 ± 0.08 b	183.17 ± 7.95 b	4.56 ± 0.18 ab
Po	26.20 ± 0.42 a	1.94 ± 0.16 a	206.56 ± 8.19 a	4.87 ± 0.19 a
NaCl	16.81 ± 0.38 d	1.51 ± 0.13 c	125.16 ± 11.35 d	3.74 ± 0.24 c
NaCl + Po	20.58 ± 0.17 c	1.72 ± 0.07 b	140.49 ± 6.51 c	4.26 ± 0.11 b

Data (mean ± SD, *n* = 5) followed by different letters in the column indicate significant differences (*p* < 0.05) between treatments.

**Table 6 plants-13-00805-t006:** Effects of AMF on leaf fluorescence parameters of cotton seedlings under salt stress.

Treatments	φPSII	Fv′/Fm′	qP	NPQ
CK	0.63 ± 0.04 b	0.86 ± 0.03 b	0.97 ± 0.08 b	0.67 ± 0.03 b
Po	0.83 ± 0.07 a	0.98 ± 0.06 a	1.22 ± 0.12 a	0.44 ± 0.01 c
NaCl	0.54 ± 0.03 c	0.77 ± 0.04 c	0.79 ± 0.04 c	0.88 ± 0.02 a
NaCl + Po	0.62 ± 0.05 b	0.84 ± 0.05 b	0.96 ± 0.07 b	0.68 ± 0.04 b

Data (mean ± SD, *n* = 5) followed by different letters in the column indicate significant differences (*p* < 0.05) between treatments.

**Table 7 plants-13-00805-t007:** Effects of AMF on concentrations of root endogenous hormones of cotton seedlings under salt stress.

Treatments	IAA (ng/g FW)	BR (ng/g FW)	GA (ng/g FW)	ABA (ng/g FW)
CK	61.13 ± 5.28 bc	2.41 ± 0.22 b	6.21 ± 0.36 a	5.84 ± 0.46 b
Po	72.22 ± 3.79 a	2.83 ± 0.12 a	6.15 ± 0.42 a	5.22 ± 0.22 b
NaCl	54.52 ± 4.21 c	1.67 ± 0.12 d	6.11 ± 0.51 a	6.98 ± 0.42 a
NaCl + Po	65.01 ± 3.44 b	2.09 ± 0.19 c	6.22 ± 0.51 a	5.88 ± 0.35 b

Data (mean ± SD, *n* = 5) followed by different letters in the column indicate significant differences (*p* < 0.05) between treatments.

**Table 8 plants-13-00805-t008:** Effects of AMF on relative water content, water saturation deficit, chlorophyll and flavonoid levels, and nitrogen balance in leaves of cotton seedlings under salt stress.

Treatments	Relative Water Content (RWC, %)	Water Saturation Deficit (WSD, %)	Chlorophyll (ug/cm^2^)	Flavonoid (ug/g)	Nitrogen Balance (g/cm^2^)
CK	62.96 ± 2.53 b	37.04 ± 2.98 a	31.43 ± 1.91 a	0.043 ± 0.003 a	732.13 ± 35.73 a
Po	65.87 ± 2.82 a	37.13 ± 2.82 a	32.04 ± 1.29 a	0.044 ± 0.004 a	755.51 ± 29.06 a
NaCl	60.10 ± 3.17 c	29.90 ± 1.98 c	26.47 ± 2.08 b	0.048 ± 0.002 a	549.76 ± 24.29 b
NaCl + Po	64.98 ± 3.23 a	36.01 ± 2.11 ab	32.03 ± 2.49 a	0.045 ± 0.003 a	748.83 ± 54.08 a

Data (mean ± SD, *n* = 5) followed by different letters in the column indicate significant differences (*p* < 0.05) between treatments.

## Data Availability

The data that support the findings of this study are available on request from the corresponding author.

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
