# Peer review of "Mycorrhizas Affect Physiological Performance, Antioxidant System, Photosynthesis, Endogenous Hormones, and Water Content in Cotton under Salt Stress"

_plants, 2024, doi:10.3390/plants13060805_

Round 1
Reviewer 1 Report
Comments and Suggestions for Authors
The manuscript entitled “Mycorrhizas affect physiological performance, antioxidant system, photosynthesis, endogenous hormones, and water content in cotton under salt stress” is a manuscript similar to many others in the same field. There is no brand new information since most part of these data here referred have been previously published more than 20-30 years ago.
The authors intends to show the relevance of this work since it is performed with cotton plants. However, there is nothing real new from the agricultural or physiological point of view. From the agricultural or ecological point of view there is no explanation on how these data will serve the economic exploitation. From the physiological point of view there is no mechanism solved. How AMF interact with cotton plants in order to improve physiological plant responses? What are the causes of salt stress: Na or Cl?
Instead at the end of your introduction you wrote.” In this study, we mainly focus on the impact and the mechanization of salinity on cotton from plant physiological phenotype”
…. What do you mean with this? What is your aim and your work hypothesis?
Under this situation I don’t think the manuscript able to be published unless the authors explore data in a different form.
For your use and convenience:
Avoid the use of siglas - POD, CAT, SOD, …- in the introduction without defining what it is, independently of the frequency of their use, and the normal use from physiological researchers
M&M
Independently of the data collected from previous authors it would be interesting to justify the option of NaCl concentration. And why only one concentration? Is it a common concentration in soil? Why did you use Paraglomus occultum? Is it a normal AMF in cotton plants or did you use just because you got it from the Root Biology Institute?
Under field conditions what AMF species are in soil of cotton plants and in the plant roots?
You said that each treatment was replicated 5 times, it is Ok, but what how many plants per treatment ? And how many plants per pot?
You have analysed growth parameters not phenotypic assay.
What kind of solution did you use for irrigation? What kind of growth regulators did you add and why?
At the beginning you said that “Each treatment was replicated five times” but at the end part of analysis of data you said all experiments were repeated four times…..
You said in the results
“Soil mycorrhizal fungal hyphae and root mycorrhizal colonization and were found in the AMF-treated seedlings, but not in the non-AMF treated seedlings, regardless of salt
stress types (Figure. 1 and Table 1). Hence, the roots of cotton established a symbiotic relationship with Paraglomus occultum (Po)”. One thing is a established symbiotic relationship, other is a normal symbiotic relationship. Besides under field conditions there is not only one AMF species growing in the roots. So, the information you gave only means that your soil/sand sterilisation has been efficient.
Concerning the data of plant physiological responses you only report the data no interpretation at all.
Author Response
Dear Editor and Reviewers:
We are thankful to the reviewers and editor for comments on the manuscript (plants-2772146 - Mycorrhizas affect physiological performance, antioxidant system, photosynthesis, endogenous hormones, and water content in cotton under salt stress) and helpful suggestions to improve the quality of our manuscript. Based on review comments and suggestions, the paper was carefully revised. Our responses to the comments are listed on the following pages. The revised manuscript has been formatted according to plants, and the text and English have been carefully checked. All corrected and added parts in the revised manuscript have been marked up using the “Track Changes” function. Also, we used English language editing by MDPI in order to meet the requirements of this journal.
Thank you for your attention and consideration of our work and we look forward to your response. Welcome to keep in touch with us, if there are any questions about this manuscript.
Kind regards.
Yours sincerely,
Dejian Zhang
Responses to reviewer 1:
The manuscript entitled “Mycorrhizas affect physiological performance, antioxidant system, photosynthesis, endogenous hormones, and water content in cotton under salt stress” is a manuscript similar to many others in the same field. There is no brand new information since most part of these data here referred have been previously published more than 20-30 years ago.
The authors intends to show the relevance of this work since it is performed with cotton plants. However, there is nothing real new from the agricultural or physiological point of view. From the agricultural or ecological point of view there is no explanation on how these data will serve the economic exploitation. From the physiological point of view there is no mechanism solved. How AMF interact with cotton plants in order to improve physiological plant responses? What are the causes of salt stress: Na or Cl?
Response: Thanks for your comments. Indeed, there has been much research on the mechanism of AMF response to salt stress in plants. However, whether AMF can also improve the salt resistance of cotton and the related mechanism have not been studied deeply. So, we mainly focus on the impact and the mechanization of salinity on cotton from antioxidant enzyme system, photosynthetic, respiratory metabolism, and phytohormone. Aim to find out the responses of cotton to salt stress and the adaptation of salt stress, and to provide more suggestions for improving cotton yield especially in saline-alkali soil. Furthermore, from the agricultural or ecological point of view there is really no explanation on how these data will serve the economic exploitation. So, we deleted “economic value” in last sentence of 1. Introduction.
Instead at the end of your introduction you wrote.” In this study, we mainly focus on the impact and the mechanization of salinity on cotton from plant physiological phenotype”
…. What do you mean with this? What is your aim and your work hypothesis?
Response: Thanks. Plant physiological phenotype in this study mean plant growth performance of cotton seedlings, such as plant height, stem diameter, leaves number, root system architecture and so on. The question you raised is very important to us. The physiological phenotype is not exact enough in this study. So, we have changed it to growth performance. Our aim is find out the responses of cotton to salt stress and the adaptation of salt stress. Our work hypothesis is that AMF may improve salt tolerance of cotton by antioxidant enzyme system, photosynthetic, phytohormone, and respiratory metabolism.
For your use and convenience:
Avoid the use of siglas - POD, CAT, SOD, …- in the introduction without defining what it is, independently of the frequency of their use, and the normal use from physiological researchers
Response: Thanks for your good suggestion. We have added the definition of POD, SOD, CAT, GR, APX, ASA, MDA in 1. Introduction.
M&M
Independently of the data collected from previous authors it would be interesting to justify the option of NaCl concentration. And why only one concentration? Is it a common concentration in soil?
Response: Thanks. In fact, we did a concentration gradient test in the early stage, through which we learned that the growth and development of cotton were significantly inhibited under 150 mmol L-1 NaCl. It is a common concentration in cotton cultivation soil.
Why did you use Paraglomus occultum? Is it a normal AMF in cotton plants or did you use just because you got it from the Root Biology Institute? Under field conditions what AMF species are in soil of cotton plants and in the plant roots?
Response: Thanks. Also, we did preliminary experiment (8 AMFs treated on cotton) in the early stage, and we learned that the growth of cotton were significantly increased when treated by Paraglomus occultum. Your second question is a very good one. We next plan to isolate and identify indigenous AMF species from cotton roots grown in the field. So, yes, we got Paraglomus occultum from the Root Biology Institute.
You said that each treatment was replicated 5 times, it is Ok, but what how many plants per treatment ? And how many plants per pot?
Response: Thanks. Each treatment was replicated 5 times, and each replicate had 5 pots, and each pot had 9 seedlings. So that each treatment had 225 seedlings. We have changed and checked it in 4.1. Experimental design.
You have analysed growth parameters not phenotypic assay.
What kind of solution did you use for irrigation? What kind of growth regulators did you add and why?
Response: Thanks. We have deleted phenotypic assay and checked it in 4.3. Variable determinations. We used the hoagland solution for irrigation, not growth regulators. We admit we made a mistake here. Thank you very much for pointing out our mistake. We have changed the growth regulators to hoagland solution in 4.2. Plant culture.
At the beginning you said that “Each treatment was replicated five times” but at the end part of analysis of data you said all experiments were repeated four times…..
Response: Thanks. We admit we made a mistake here. Thank you very much for pointing out our mistake. We have changed it to 5 times full manuscript.
You said in the results
“Soil mycorrhizal fungal hyphae and root mycorrhizal colonization and were found in the AMF-treated seedlings, but not in the non-AMF treated seedlings, regardless of salt
stress types (Figure. 1 and Table 1). Hence, the roots of cotton established a symbiotic relationship with Paraglomus occultum (Po)”. One thing is a established symbiotic relationship, other is a normal symbiotic relationship. Besides under field conditions there is not only one AMF species growing in the roots. So, the information you gave only means that your soil/sand sterilisation has been efficient.
Response: Thanks. The fact is that one thing is a established symbiotic relationship (the treatments of CK and NaCl), other is an non-established symbiotic relationship (the treatments of Po and NaCl+Po). You are wright, we did soil/sand sterilisation were efficient.
Concerning the data of plant physiological responses you only report the data no interpretation at all.
Response: Thanks. We discussed the growth parameters data in the 3. Discussion section.

Reviewer 2 Report
Comments and Suggestions for Authors
Lines 107-108: Mycorrhizal colonization was 77.4% under Po and not under CK (see Table 1.
line 379: Four treatments include CK, CK+NaCl, Po and Po+NaCl but all tables show CK, Po, NaCl and Nacl+Po. Where is CK+NaCl on the tables? Treatments under materials and methods must match those on the tables. Note that a treatment of CK+NaCl is different from CK and NaCl as separate treatments. The tables report them as separate treatments as opposed to the methodology of materials and methods.
Line136: What is root projected area? This must be clarified as it is impossible to project the extent of root growth in the soil.
Lines 182-184: The computation of Pn, Gs, Ci and Tr percentages do not seem correct. Check to ensure that the computations in parenthesis are correct.
Comments on the Quality of English LanguageA major problem with this excellent study is communication. This manuscript requires extensive editing of the English language. I could have edited it if I had time but for now, I will list the lines that require thorough editing and rephrasing of the language.
The following lines need grammar correction:
20, 22, 27-28, 32-35, 46-47, 49-54, 68-72, 86-88, 93-94, 97-101, 132-132, 138, 139-140, 142 (Figure 2. Whole plant morphology of cotton seedlings treated with AMF under salt stress), 135-138, 167-168 (pared with non-AMF inoculation, the activities of SOD, POD, and CAT in roots of cotton treated with AMF were significantly increased regardless of salt stress or not (Table 4), 170-173, 194-196, 199, 206-207, 211, 215, 222 (2.8. Effects of salt stress and AMF on leaf water content, chlorophyll and nitrogen balance), 223, 226, 239-240, 245-249, 261-263, 270-271, 278-279, 289-297, 302-303, 309, 324, 326-327, 330, 334-335,352, 356, 377, 385, 386-398 (these two paragraphs need to be rewritten for clarity), 405, 409, 414, 423, 428-429 (RWC and WSD are calculated using the following formula. Where is the formula?), 430-436 (the entire paragraph is difficult to comprehend), 448 (use of the word "observably" is strange. If the result is significant, why not use "significantly" How can you observe root dry weight? To observe means to watch or see with the eyes), 453, 455-456, 459-463, 466, and 469.
Again, this is an excellent manuscript but it requires extensive grammar editing including rephrasing of many sentences.
Author Response
Dear Editor and Reviewers:
We are thankful to the reviewers and editor for comments on the manuscript (plants-2772146 - Mycorrhizas affect physiological performance, antioxidant system, photosynthesis, endogenous hormones, and water content in cotton under salt stress) and helpful suggestions to improve the quality of our manuscript. Based on review comments and suggestions, the paper was carefully revised. Our responses to the comments are listed on the following pages. The revised manuscript has been formatted according to plants, and the text and English have been carefully checked. All corrected and added parts in the revised manuscript have been marked up using the “Track Changes” function. Also, we used English language editing by MDPI in order to meet the requirements of this journal.
Thank you for your attention and consideration of our work and we look forward to your response. Welcome to keep in touch with us, if there are any questions about this manuscript.
Kind regards.
Yours sincerely,
Dejian Zhang
Lines 107-108: Mycorrhizal colonization was 77.4% under Po and not under CK (see Table 1.)
Response: Thank you for your comments of this article. The sand was collected from the Yangtze River side and autoclaved at 0.11 MPa, 121 °C for 2 h to eliminate spores of indigenous arbuscular mycorrhizal fungi in the sand. (see 4.2. Plant culture). So Mycorrhizal colonization was 0.00% under CK.
line 379: Four treatments include CK, CK+NaCl, Po and Po+NaCl but all tables show CK, Po, NaCl and Nacl+Po. Where is CK+NaCl on the tables? Treatments under materials and methods must match those on the tables. Note that a treatment of CK+NaCl is different from CK and NaCl as separate treatments. The tables report them as separate treatments as opposed to the methodology of materials and methods.
Response: The problem you refer to is very important. Thanks very much. You are right that a treatment of CK+NaCl is different from CK and NaCl as separate treatments. In fact, we only have 4 treatments that CK, Po, NaCl, and Nacl+Po. We have made changes in line 379. The tables report them bring into correspondence with the methodology of materials and methods. Thanks again.
Line136: What is root projected area? This must be clarified as it is impossible to project the extent of root growth in the soil.
Response: Root projected area refers to the surface area of a cotton's root system as it appears when projected onto a two-dimensional plane. We carefully removed the whole cotton’s root system and then used an Epson Perfection V700 Photo Dual Lens System (J221A, Indonesia) to project the extent of root growth in the soil. And the WinRHIZO software (2007b, Regent Instruments Inc., Quebec, Canada) was used for data analysis.
Lines 182-184: The computation of Pn, Gs, Ci and Tr percentages do not seem correct. Check to ensure that the computations in parenthesis are correct.
Response: Thank you very much for your meticulous review. Your meticulous attitude is worth learning from. We have recalculated and checked these percentages carefully.
A major problem with this excellent study is communication. This manuscript requires extensive editing of the English language. I could have edited it if I had time but for now, I will list the lines that require thorough editing and rephrasing of the language.
The following lines need grammar correction:
20, 22, 27-28, 32-35, 46-47, 49-54, 68-72, 86-88, 93-94, 97-101, 132-132, 138, 139-140, 142 (Figure 2. Whole plant morphology of cotton seedlings treated with AMF under salt stress), 135-138, 167-168 (pared with non-AMF inoculation, the activities of SOD, POD, and CAT in roots of cotton treated with AMF were significantly increased regardless of salt stress or not (Table 4), 170-173, 194-196, 199, 206-207, 211, 215, 222 (2.8. Effects of salt stress and AMF on leaf water content, chlorophyll and nitrogen balance), 223, 226, 239-240, 245-249, 261-263, 270-271, 278-279, 289-297, 302-303, 309, 324, 326-327, 330, 334-335,352, 356, 377, 385, 386-398 (these two paragraphs need to be rewritten for clarity), 405, 409, 414, 423, 428-429 (RWC and WSD are calculated using the following formula. Where is the formula?), 430-436 (the entire paragraph is difficult to comprehend), 448 (use of the word "observably" is strange. If the result is significant, why not use "significantly" How can you observe root dry weight? To observe means to watch or see with the eyes), 453, 455-456, 459-463, 466, and 469.
Again, this is an excellent manuscript but it requires extensive grammar editing including rephrasing of many sentences.
Response: Thanks very much for your a lot of review works. We have made modifications one by one according to your modification suggestions. Also, we used English language editing by MDPI in order to meet the requirements of this journal.

Round 2
Reviewer 1 Report
Comments and Suggestions for Authors
Dear Authors
I regret to say that you did not provide a clear and sufficient response to my comments.
- Work justification: you only added “However, the potential of AMF to improve salt resistance in cotton and the related mechanisms have not been extensively studied….Additionally, we aim to identify the responses of cotton to salt stress, understand its adaptation mechanisms, and provide more suggestions for improving cotton yield”. This could be good IF you provided truly mechanisms.
- Conclusion: “In summary, mycorrhizal cotton seedlings may exhibit mechanisms involving root system architecture, the antioxidant system, photosynthesis, leaf fluorescence, endogenous hormones, water content, and nitrogen balance that increase their resistance to saline–alkali environments. Our study provide a theoretical basis for further exploring the application of AMF to heighten the salt tolerance of cotton.” This is nothing of mechanisms, only a repetition of what you have written in introduction, results and discussion. Please provide what were the novelties you did approach in this study.
- Concerning M&M section you also provided more explanations in your direct response than in amendments in the text.
Author Response
Dear Editor and Reviewer:
We are thankful to the reviewer and editor again for comments on the manuscript (plants-2772146 - Mycorrhizas affect physiological performance, antioxidant system, photosynthesis, endogenous hormones, and water content in cotton under salt stress) and helpful suggestions to improve the quality of our manuscript. I am sorry for not provide a clear and sufficient response to your comments. So, we have made careful revisions to the MS again. Our responses to the comments are listed on the following pages. All corrected and added parts in the revised manuscript have been marked up using the “Track Changes” function.
Thank you for your attention and consideration of our work and we look forward to your response. Welcome to keep in touch with us, if there are any questions about this MS.
Kind regards.
Yours sincerely,
Dejian Zhang
Responses to reviewer:
- Work justification: you only added “However, the potential of AMF to improve salt resistance in cotton and the related mechanisms have not been extensively studied….Additionally, we aim to identify the responses of cotton to salt stress, understand its adaptation mechanisms, and provide more suggestions for improving cotton yield”. This could be good IF you provided truly mechanisms.
Response: Thanks for your comments. It is true that we have not proposed a truly mechanisms for enhancing salt tolerance of cotton. Exactly, there has been a lot of research on AMF improving salt tolerance in crops. However, we confirm that Paraglomus occultum can significantly enhance the salt tolerance of cotton through preliminary screening of different AMFs and explained the reason in this study. In the future, we plan to propagate Paraglomus occultum on a large scale and apply it in the field to improve the damage of salt stress on cotton. Thanks for your comments again.
- Conclusion: “In summary, mycorrhizal cotton seedlings may exhibit mechanisms involving root system architecture, the antioxidant system, photosynthesis, leaf fluorescence, endogenous hormones, water content, and nitrogen balance that increase their resistance to saline–alkali environments. Our study provide a theoretical basis for further exploring the application of AMF to heighten the salt tolerance of cotton.” This is nothing of mechanisms, only a repetition of what you have written in introduction, results and discussion. Please provide what were the novelties you did approach in this study.
Response: Thanks. To be honest, we're not provide novel insights in this study. However, we provide a more comprehensive analysis from root system architecture, the antioxidant system, photosynthesis, leaf fluorescence, endogenous hormones, water content, and nitrogen balance.
- Concerning M&M section you also provided more explanations in your direct response than in amendments in the text.
Response: Thanks for your good suggestion. We have added these more explanations (in my direct response to you) to amendments in the revised MS.

Reviewer 2 Report
Comments and Suggestions for Authors
The effort by the authors to improve the language of the manuscripts should be commended. This journal is widely read throughout the world. The authors should endeavor to work on the English language presentation in any future article. Language made the paper extremely difficult to read and comprehend in the first round of the peer review process.
Author Response
Dear Editor and Reviewer:
We are very thankful to the reviewer and editor for comments on the manuscript (plants-2772146 - Mycorrhizas affect physiological performance, antioxidant system, photosynthesis, endogenous hormones, and water content in cotton under salt stress) and helpful suggestions to improve the quality of our manuscript.
We are also grateful for your approval of this MS. Thanks again.
Kind regards.
Yours sincerely,
Dejian Zhang
Round 3
Reviewer 1 Report
Comments and Suggestions for Authors
You should avoid repetitions in conclusion. Be strict with your message emphasising the need of more studies and the potential application of field inoculation
Author Response
Dear Editor and Reviewer:
We are thankful to the reviewer and editor again for comments on the manuscript (plants-2772146 - Mycorrhizas affect physiological performance, antioxidant system, photosynthesis, endogenous hormones, and water content in cotton under salt stress) and helpful suggestions to improve the quality of our manuscript. We have condensed the conclusion section so as avoid repetitions in conclusion. Also, we have added “However, more studies and the potential application of field inoculation are needed” in conclusion section.
Thank you again for your very good suggestions.
Kind regards.
Yours sincerely,
Dejian Zhang
